# EGAIN: Enhanced Generative Adversarial Networks for Imputing Missing Values

## Abstract

Missing values pose a challenge in predictive analysis specially in big data because most models depend on complete datasets to estimate functional relationships between variables. Generative Adversarial Imputation Networks (GAIN) are among the most reliable methods to predict and impute missing values. This research introduces Enhanced Generative Adversarial Networks (EGAIN), which address the GAIN convergence issue, introduce new functionality to the GAIN process, and significantly improve its performance.

## 1 Introduction

Missing values are a common issue in predictive analysis, as most models require complete data to estimate functional relationships among existing variables. There are two main approaches to address missing values in datasets: (1) case deletion, where an entire row of data is removed if it contains at least one missing value, and (2) missing value imputation, where plausible values are estimated and filled in for the missing data. Each method has its drawbacks. Case deletion can significantly reduce the number of samples for predictive analysis, particularly in datasets with a high proportion of missing values, thereby reducing the power of estimations. On the other hand, imputing missing values allows partial information from rows with missing values to be used but may lead to biased results if improper imputations are applied.

Moreover, there are three types of missing values: Missing Completely At Random (MCAR), Missing At Random (MAR), and Missing Not At Random (MNAR) (Rubin, 1976). In MCAR, data is missing purely by chance, e.g., when a survey response is lost because the respondent did not see the question. In this scenario, the missingness is not related to any observed or unobserved data. MAR occurs when the missingness is related to some observed data but not the missing values themselves, e.g., older participants are more likely to skip an income question. In this case, the missing data can be accounted for by other known variables, such as age. MNAR arises when the reason for missingness depends on the unobserved missing data itself, e.g., people with high incomes may choose to hide their salary, resulting in missing income information.

Median imputation is a simple and widely used baseline method for handling missing data. This approach replaces missing values with the median of the observed values for a given variable, which helps preserve the central tendency while being robust to outliers. Despite its simplicity, median imputation may underestimate variability and distort relationships among variables. Nonetheless, Median imputation is often used as a benchmark in evaluating more sophisticated imputation techniques (Little & Rubin , 2019).

There are many advanced imputation methods designed to reliably handle various types of missing data. Among them is Multiple Imputation by Chained Equations (MICE), which iteratively models each variable with missing data as a function of the others, using regression techniques (Van Buuren and Groothuis-Oudshoorn, 2011). MICE is particularly well-suited for datasets containing both numerical and categorical variables and is praised for its ability to preserve the multivariate relationships among variables. However, its iterative nature makes it computationally intensive, and it may perform poorly when the relationships between variables are non-linear (White et al., 2011). Additionally, MICE can struggle with categorical variables, sometimes producing implausible imputations outside the allowable category set, especially when the imputation model is not properly specified (Azur et al., 2011). An alternative method, MissForest, uses random forests to impute missing values and is capable of capturing complex, non-linear interactions

between variables (Stekhoven and Buhlmann, 2011). It performs well on mixed-type data and requires minimal tuning. Despite these advantages, MissForest is computationally expensive and may scale poorly with large datasets. Moreover, its accuracy declines when the proportion of missing data is high, and it may introduce bias in datasets characterized by extreme values or highly skewed distributions (Waljee et al., 2013).

Generative Adversarial Imputation Networks (GAIN) is a deep learning-based approach for missing value imputation that models the distribution of observed data to impute missing values (Yoon et al., 2018). GAIN handles both numerical and categorical variables and performs well even when the data distribution is imbalanced or skewed (Dong et al., 2021). However, it is sensitive to hyperparameter selection, and suffers from encoding issues and reduced accuracy for multi-class variables. In addition to GAIN, MissForest, MICE and baseline methods like median imputation, several other missing value imputation techniques have been developed, each with varying strengths and weaknesses. For a comprehensive review of models for handling missing data, see (Zhou et al., 2024).

Several researchers have provided performance comparisons between MICE, MissForest, and GAIN on benchmark datasets (Dong et al., 2021; Sun et al., 2023; Shahbazian and Greco, 2023). The results indicate that the performance of these models depends heavily on factors such as the type (numerical/categorical) and number of variables, variable skewness, number of cases, the percentage and type of missing values. GAIN has demonstrated superior performance over MICE and MissForest in many areas, including speed and handling high missing rates.

Several variations of GAIN have been proposed since its introduction in 2018. Among them are: LFM-D2GAIN, which integrates a latent factor model (LFM) for coherent training and reduced reconstruction error, along with a dual-discriminator (D2) to capture multi-modal data distribution (Shen et al., 2022). Their approach improves training coherence, reduces reconstruction error, and captures multi-modal data distributions to prevent mode collapse. GAGIN integrates global and local imputation networks with an imputation guider model to address local homogeneity and improve prediction performance (Wang et al., 2022). Experimental results show GAGIN significantly outperforms state-of-the-art and traditional methods, achieving up to 17.3% and 24.1% improvements in RMSE on numeric and image datasets, respectively. ClueGAIN incorporates transfer learning to improve imputation accuracy in datasets with high missing rates (Zhao, 2023). Experimental results on Cancer Patients DNA Sequence dataset shows on average a 11.01% decrease in RMSE across 60%-90% missing rates. ccGAIN enhances imputation accuracy by conditioning imputation on observed and annotated values in clinical data with high missing rates (Bernardini et al., 2023). They show that ccGAN significantly outperforms existing methods including Mice and MissForest, achieving up to 19.79% improvement in imputation accuracy and enhanced robustness across varying missingness levels on real-world EHR datasets. LWGAIN integrates the Wasserstein distance in the loss function and incorporates labeled inputs into the generator, improving imputation performance by reducing RMSE on the Kansas logging dataset therby enabling effective lithology identification (Qian et al., 2024). MGAIN addresses gradient vanishing and mode collapse using least squares loss and dual discriminators (Qin et al., 2024). Experimental results show that MGAIN reduces RMSE by 21.66% compared to baseline models across several benchmark datasets. However, the code for these GAIN variants is not publicly available at the time of this study, limiting reproducibility and broader adoption of their approaches.

Other GAIN variants whose codes are publicly available include scGAIN for imputing missing gene expressions in single-cell RNA sequencing data with competitive performance and improved biological relevance in both simulated and real datasets (Gunday et al., 2019); CGAIN, a class-aware imputation method based on Conditional GAIN that models class-specific data distributions to improve missing value estimation(Awan et al., 2020); COGAIN, that adds a weighting losses for mixed data types to efficiently impute missing data in large real-world clinical datasets under high missingness rates and with skewed or imbalanced variables Dong et al. (2021).

Despite the many variations built on GAIN, its application still relies on the outdated `TensorFlow 1.x` Application Programming Interface (API) and several nonstandard user-defined functions for scaling, sampling, and network initiation. Moreover, its deep networks that are considered the center of the process are simple deep neural networks, unable to discover the spatial relationships in the input data. Crucially, GAIN imple-

mentation is very sensitive to the choice of hyperparameters and often exhibits convergence issues (Kazemi & Meidani , 2020; Qin et al., 2024), especially when missing data is limited to a small number of variables, a characteristic prevalent in most real-life datasets. Indeed, Sun et al. (2023) highlight the lack of standardized software for the GAIN method.

To address these limitations, we introduce Enhanced Generative Adversarial Imputation Networks (EGAIN), a modernized and robust extension of GAIN. EGAIN improves convergence stability through the integration of model checkpointing and diagnostic visualizations, making it easier to monitor training progress and identify suboptimal hyperparameters. Unlike the original GAIN, which is implemented using the deprecated `TensorFlow 1.x` API and depends on nonstandard functions, EGAIN is built from the ground up in `TensorFlow 2`, adhering to modern software practices for greater transparency, reproducibility, and ease of use. Importantly, EGAIN replaces traditional fully connected layers with convolutional layers, which enable the model to capture local structural dependencies among features; an approach shown to improve imputation accuracy in structured tabular data. EGAIN also introduces a revised input formatting strategy that stacks data and mask matrices as separate channels, allowing for more expressive feature interactions. The source code, sample datasets, and installation instructions are available as a user-friendly `Python 3.x` package on `PyPi` and `GitHub`, facilitating immediate integration into analytical workflows. Together, these enhancements make EGAIN a more stable, interpretable, and effective solution for missing data imputation in real-world applications.

## 2 Model Description

The Generative Adversarial Imputation Network (GAIN), introduced by Yoon et al. (2018) formulates the imputation of missing values as a supervised learning problem using a generative adversarial framework. It employs a generator a generator ($G$) and a discriminator ($D$) in a competitive setting inspired by Generative Adversarial Networks (GANs). The core idea of GAIN is to generate plausible imputations for missing values using a generator, denoted as

$$\hat{X} = G\left(\tilde{X}, M, Z\right), \tag{1}$$

where $\tilde{X}$ is the data array whose missing values are replaced with zero, $M$ is the binary mask array whose values are 1 for observed data, and 0 for missing, and $Z$ is random noise applied only to missing value arrays. After the generator ($G$) imputes the missing values, the discriminator ($D$) attempts to distinguish real values from imputed ones by outputting a probability array that indicates the chance of each component being real, using:

$$D\left(\hat{X}, H\right), \tag{2}$$

where $\hat{X}$ is the output of the generator, and $H$ is the hint array that provides partial information about which values are missing. The generator and discriminator networks are trained over a large number of iterations, while improving their performance by reducing competing loss functions. The discriminator is trained to maximize classification (real/imputed) accuracy by minimizing the following binary cross entropy loss function:

$$\mathcal{L}_D = -\mathbb{E}_{\hat{X},M,H}\left[M \log D(\hat{X}, H) + (1-M)\log(1 - D(\hat{X}, H))\right]. \tag{3}$$

The generator is trained to minimize the discriminator's ability to differentiate real values from imputed ones, with the following loss function:

$$\mathcal{L}_G = -\mathbb{E}_{\tilde{X},M}\left[(1-M)\log D(\hat{X}, H)\right]. \tag{4}$$

This loss function is only applied to imputed missing ($m_i = 0$) and penalizes the generator ($G$) if the discriminator ($D$) performs well by correctly outputting low chances. To encourage the generator to produce realistic values that deceive the discriminator, a reconstruction loss is added to the generator:

$$\mathcal{L}_M = \begin{cases} \sum m_i(x_i - \hat{x}_i)^2, & \text{if } x_i \text{ is continuous,} \\ -\sum m_i(x_i \log(\hat{x}_i)), & \text{if } x_i \text{ is binary.} \end{cases} \tag{5}$$

This loss function is only applied to observed values ($m_i = 1$). Therefore, the total generator loss becomes:

$$\mathcal{L}_G^{total} = \mathcal{L}_G + \alpha\mathcal{L}_M,\tag{6}$$

where $\alpha$ is a hyperparameter that controls the contribution of the reconstruction loss to the overall objective. The competition between the generator and the discriminator drives the generator to produce high-quality imputations that are indistinguishable from real data. It is important to note that only the continuous reconstruction loss has been used in the GAIN implementation and its successors.

The following are a series of enhancements that has been applied to the GAIN implementation, driving the EGAIN with improved performance:

- **Modernized Framework with TensorFlow 2 API**
  EGAIN is implemented using the TensorFlow 2.x API, replacing the outdated TensorFlow 1.x-based GAIN code. This modernization ensures compatibility with current tools and libraries, improves code readability, and aligns with best practices in deep learning development.

- **Use of Standard Built-in Functions**
  The original GAIN implementation relied on multiple custom utility functions for scaling, sampling, and model setup, many of which lacked documentation or standard structure. EGAIN replaces these with standardized, well-tested TensorFlow/Keras functions, improving maintainability, reproducibility, and reducing debugging overhead.

- **Improved Network Architecture via Convolutional Layers**
  EGAIN replaces the fully connected dense layers in both the generator and the discriminator with 1D convolutional layers. These layers treat the input data and the missingness mask as separate channels, enabling the model to capture local dependencies and feature-level spatial patterns that are often overlooked by traditional dense architectures. This change has shown to improve imputation performance, especially in datasets with structured or correlated features.

- **Enhanced Input Formatting Strategy**
  Instead of concatenating input and mask vectors side-by-side (as done in GAIN), EGAIN stacks them as separate channels in a 2D format. This structure is particularly suited to convolutional layers and allows the network to model the interactions between observed and missing data more effectively.

- **Integrated Checkpointing and Convergence Diagnostics**
  To address GAIN's known instability and convergence issues, EGAIN incorporates checkpointing to save the best-performing model during training. Additionally, it provides visualizations of the generator and discriminator loss curves, enabling users to better monitor training behavior and adjust hyperparameters accordingly.

- **Improved Hyperparameter Tuning Support**
  EGAIN reduces the trial-and-error typically associated with tuning GAIN by offering diagnostic plots and clearer loss metrics. These tools assist users in identifying well-calibrated values for sensitive parameters such as batch size, and generator-discriminator balance ($\alpha$), resulting in more consistent training outcomes.

- **Robust, Reproducible, and User-Friendly Package**
  EGAIN is released as a fully documented, pip-installable Python package, with code and examples available on GitHub. This facilitates immediate deployment and ensures reproducibility across different environments and use cases.

## 3 Datasets and Experimental Setup

Table 1 summarizes the benchmark UCI datasets Kelly et al. (2025) used to evaluate the performance of the EGAIN model in comparison to GAIN Yoon et al. (2018) and baseline Median imputation (Little & Rubin , 2019). For each dataset, various missingness settings were considered across a range of missing value rates.

Missing values were introduced under the MCAR assumption using standard procedures following Rubin (1976), with different random seeds to generate multiple incomplete datasets.

Imputation was performed using either EGAIN or GAIN on each of the resulting incomplete datasets. Crucially, the imputation process operated without access to the original complete data, thereby emulating a realistic imputation setting. Hyperparameters for each model were selected using 5-fold cross-validation on the incomplete datasets. Following imputation, performance was assessed by comparing the imputed values to the known ground truth from the original complete data, using Root Mean Squared Error (RMSE) as the evaluation metric:

$$RMSE = \sqrt{\frac{1}{n} \sum_{t=1}^{n} (x_i - \hat{x}_i)^2}, \tag{7}$$

where $n$ is the number of missing values, $x_i$ is the true value, and $\hat{x}_i$ is the imputed value.

To ensure fair and stable comparisons, the entire process was repeated independently 25 times using consistent random seeds across both models. All variables were scaled to a $[0, 1]$ range using Min/Max scaling prior to RMSE calculation. A hint rate of 90% was used in all imputations. Batch sizes were set to 64 for small and medium datasets, and 256 for large datasets. The hyperparameter $\alpha$ was selected by monitoring the generator and discriminator loss curves to ensure comparable initial magnitudes (see Figure 2). Line charts report the mean and standard deviation of RMSE across the 25 runs. Additional implementation and hyperparameter details are provided in the supplementary materials.

Table 1: Benchmark datasets.

| Dataset | Cases (n) | Variables (d) | Description |
|---|---|---|---|
| Breast Cancer Wisconsin | 569 | 31 | 30 numerical, 1 binary categorical |
| Spambase | 4,601 | 58 | 57 numerical, 1 binary categorical |
| Letter Recognition | 20,000 | 17 | 17 categorical |
| Default of Credit Card Client | 30,000 | 24 | 14 numerical, 10 binary categorical |
| Online News Popularity | 39,797 | 59 | 45 numerical, 14 binary categorical |

## 4 Results

Figure 1 (left) and Table 3 present the RMSE performance of GAIN, EGAIN, and Median imputation on the Breast Cancer dataset under varying levels of missingness, where missing values were introduced completely at random across the 30 numerical predictors. EGAIN consistently outperformed GAIN, achieving a statistically significant ($p < 0.001$) reduction in RMSE ranging from 5.10% to 15.78%. Notably, GAIN's convergence was highly sensitive to the number of training iterations, often failing to produce results beyond an optimal threshold. Consequently, different iteration counts were required across missing rates (see supplementary material). In contrast, EGAIN demonstrated stable convergence across all settings, with performance improving steadily as the number of iterations increased. Both GAIN and EGAIN outperformed Median imputation across all levels of missingness, though the margin narrowed at higher rates. At 70% missingness, GAIN underperformed Median by 11.55%, while EGAIN maintained a 4.87% improvement over the baseline Median imputation.

The limitations of the original GAIN implementation become more pronounced when missing values are confined to a subset of columns; a scenario frequently encountered in real-world datasets. As shown in Figure 1(right) and Table 4, missing values were introduced completely at random but restricted to a randomly selected subset of the 30 numerical predictors. EGAIN demonstrated significant performance gains over GAIN, with RMSE reductions ranging from 5.25% (16 columns affected at 75% total missingness) to 32.59% (8 columns affected at 75% total missingness), all statistically significant ($p < 0.001$). In contrast, GAIN failed to produce results more frequently as the number of training iterations increased, necessitating adjustment of iteration counts across different settings. Detailed results and corresponding hyperparameter

configurations are provided in the supplementary materials. Both GAIN and EGAIN outperformed the baseline Median imputation across all tested conditions.

Figure 2, generated using EGAIN, displays the discriminator and generator loss functions across training iterations. Both the discriminator and generator improve their performance over iterations, as indicated by the decreasing loss values. Around iteration 100, the generator loss $\mathcal{L}_G$ begins to increase slightly, reflecting the discriminator's improved ability to distinguish real from imputed values, thereby pushing the generator to produce more realistic imputations. This plot also facilitates the selection of the hyperparameter $\alpha$ by enabling alignment of the initial loss scales between the discriminator and generator; in this example, $\alpha = 80$ was chosen to achieve this balance. Although training continued for 1,000 iterations, the lowest generator loss occurred around iteration 520. EGAIN automatically stores the model weights at this optimal point via checkpointing and uses them for the final imputation. This capability is absent in the original GAIN implementation, which lacks checkpointing and is more prone to overtraining; often resulting in failed imputations. Additionally, EGAIN scales the discriminator loss by a factor of 10 to emphasize its influence during training and to produce smoother, more interpretable loss curves. Importantly, EGAIN also allows training to be resumed from checkpointed weights, enabling further improvement in imputation quality across successive runs.

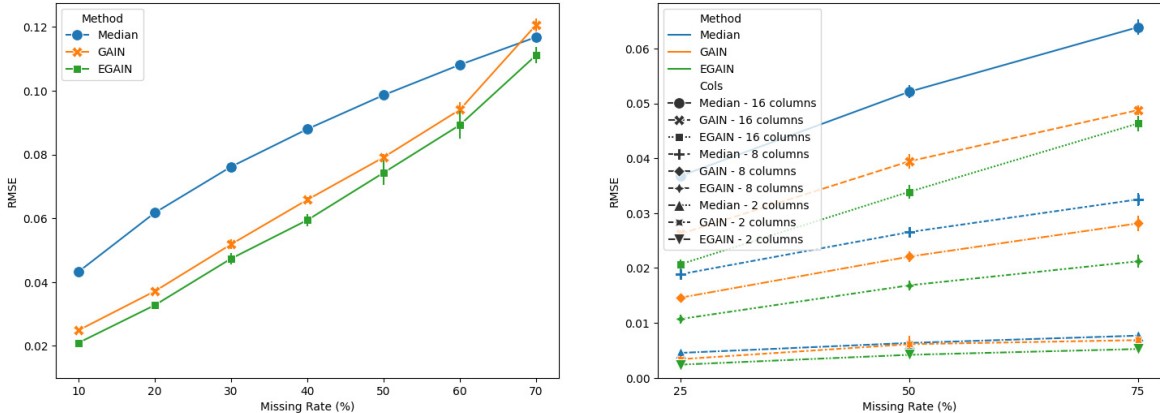

Figure 1: Performance comparison of GAIN vs EGAIN for Breast Cancer dataset: (left) MCAR from the 30 columns. (right) MCAR from randomly selected columns among 30 numerical features.

Figure 3 presents a performance comparison between GAIN, EGAIN, and Median imputation on the Spambase dataset. On the left, missing values were selected completely at random from all 57 numerical predictors at varying missing rates. Across all missing rates, EGAIN consistently outperformed both the original GAIN and the baseline Median imputation. A two-way ANOVA confirmed a statistically significant difference in performance among the three methods ($p < 0.001$). Post hoc comparisons ranked EGAIN as the top-performing imputation method, followed by GAIN and then Median. The performance gap widened notably on the right panel, where missing values were restricted to randomly selected subsets of the 57 features at various rates. As observed previously, the original GAIN frequently failed to produce results when the number of iterations surpassed an optimally chosen threshold.

Figure 4 (left) shows the RMSE performance of GAIN, EGAIN, and Median imputation on the Letter Recognition dataset, where missing values were introduced completely at random across the 16 categorical predictors. EGAIN consistently outperformed the baseline Median imputation across all missingness levels. While GAIN slightly outperformed EGAIN at lower missingness levels (by 1.09%, 2.88%, and 3.46% at 10%, 20%, and 30% respectively), EGAIN demonstrated superior robustness as missingness increased. Notably, at 70% missingness, EGAIN outperformed GAIN by 17.93% and Median by 0.94%, highlighting its effectiveness in high-missingness scenarios.

As shown in Figure 4 (right), when missing values were confined to a randomly selected subset of the 16 predictors, EGAIN achieved clear and substantial improvements over both GAIN and Median. The most

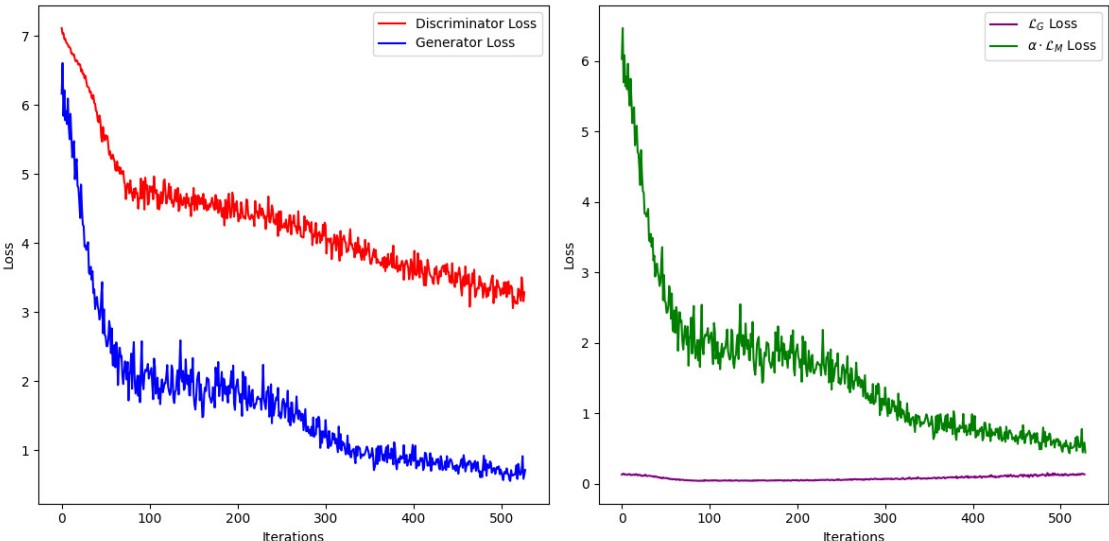

Figure 2: Progress of the loss functions throughout training in breast cancer data.

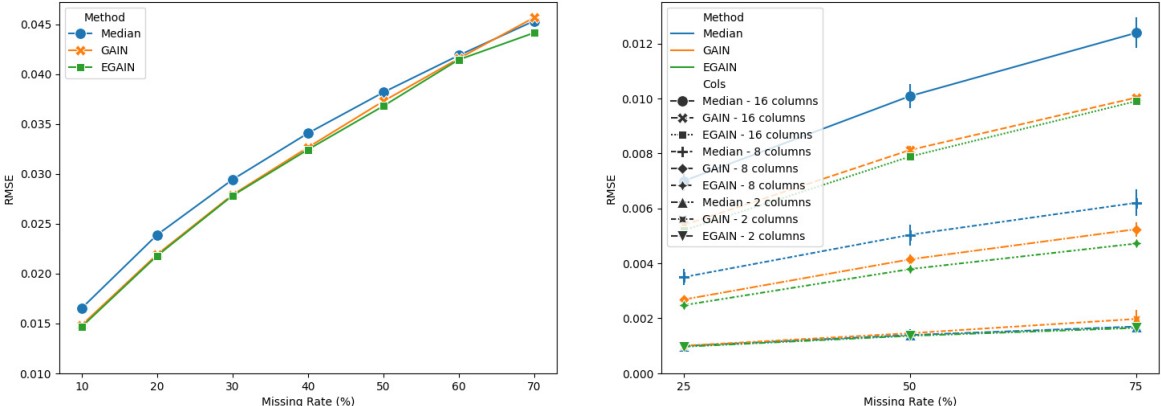

Figure 3: Performance comparison of GAIN vs EGAIN for Spambase dataset: (left) MCAR from the 57 columns. (right) MCAR from randomly selected columns.

pronounced gains were observed when only two features contained missing values and the overall missingness rate reached 75%, demonstrating EGAIN's strength in structured and sparsely missing data settings.

Performance comparisons between GAIN, EGAIN, and Median imputation on the Credit Card Client dataset are presented in Figure 5. EGAIN demonstrated statistically significant improvements over both GAIN ($p < 0.001$) and the baseline Median imputation ($p < 0.001$) across different levels of missingness. The performance gap was especially pronounced when missing values were restricted to a random subset of features, with EGAIN outperforming GAIN by 12.64% to 24.69%, and Median imputation by 16.73% to 47.80%. Notably, GAIN remained highly sensitive to the number of training iterations and frequently failed to produce results beyond the optimal threshold selected in this study; see supplementary data for details.

Figure 6 (left) presents the performance of GAIN, EGAIN, and Median imputation on the Online News Popularity dataset, where missing values were introduced completely at random across all 58 predictors. Both GAIN and EGAIN consistently outperformed the baseline Median imputation across all levels of missingness. Although GAIN slightly outperformed EGAIN at lower missingness levels, by 2.62%, 2.01%, and 0.85% at 10%, 20%, and 30% respectively, EGAIN showed increasingly superior performance as the missingness rate rose, demonstrating stronger robustness under more challenging conditions.

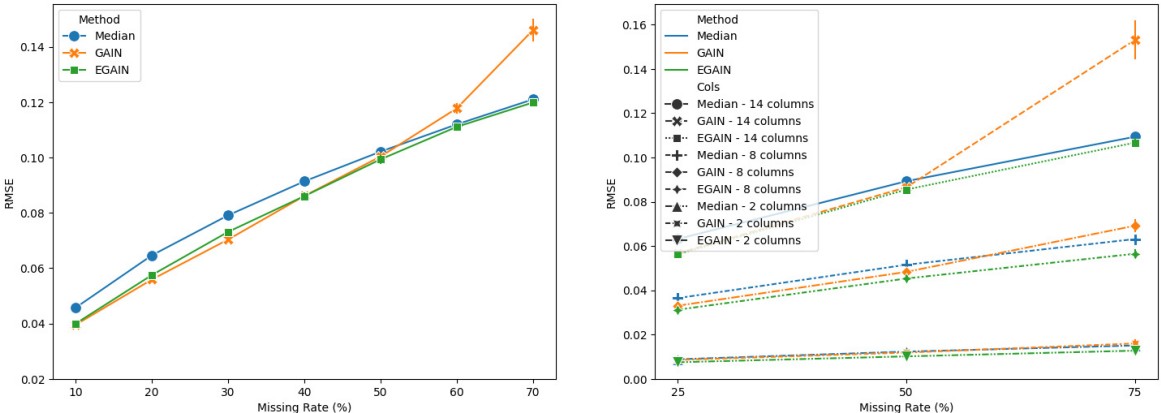

Figure 4: Performance comparison of GAIN vs EGAIN for Letter Recognition dataset: (left) MCAR from the 16 categorical columns. (right) MCAR from randomly selected columns among 16 categorical features.

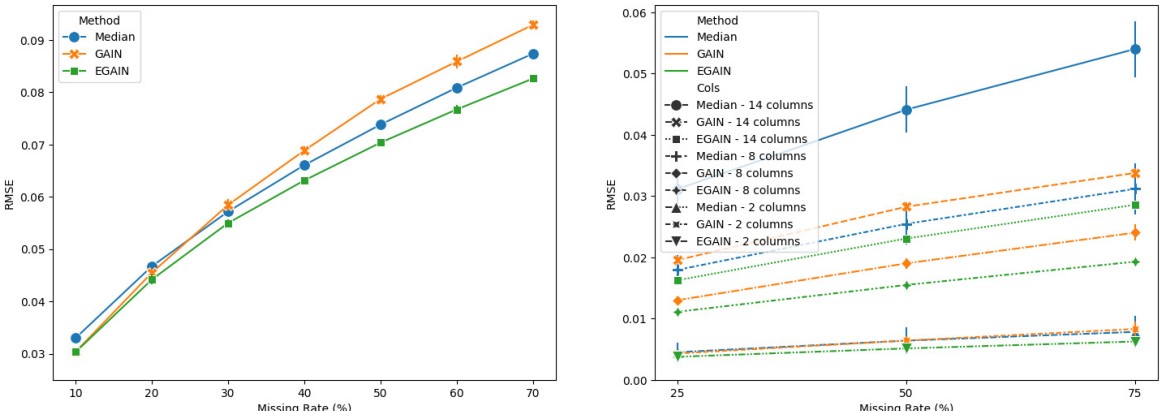

Figure 5: Performance comparison of GAIN vs EGAIN for Credit Card Client dataset: (left) MCAR from the 14 numerical columns. (right) MCAR from randomly selected columns among 14 numerical features.

As shown in Figure 6 (right), when missing values were restricted to a randomly selected subset of the 58 predictors, EGAIN delivered clear and substantial improvements over both GAIN and Median imputation. These results further highlight EGAIN's ability to handle structured missingness effectively, especially when the missingness is concentrated in fewer features.

Table 2 summarizes the overall imputation performance across all benchmark datasets. EGAIN consistently outperformed the original GAIN implementation, achieving statistically significant RMSE reductions of 3.11%, 19.85%, 15.12%, 16.48%, and 1.61% on the Breast Cancer, Spambase, Letter Recognition, Credit Card Client, and News Popularity datasets, respectively. Additionally, EGAIN consistently surpassed the baseline Median imputation method across all datasets, whereas GAIN failed to do so in both the Spambase and Letter Recognition tasks. Beyond improved accuracy, one of EGAIN's most important contributions is its resolution of a key limitation in GAIN—training instability. Across 2,000 simulation runs, GAIN failed to complete imputations in approximately 39% of cases due to convergence issues. In contrast, EGAIN completed all runs successfully. Detailed outcomes for each simulation, analysis of variance, and post hoc comparisons are provided in the supplementary materials.

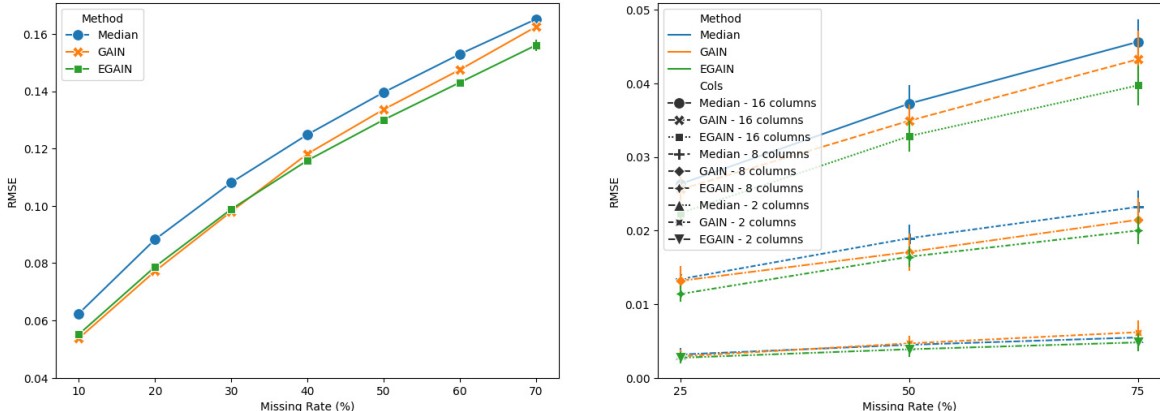

Figure 6: Performance comparison of GAIN vs EGAIN for Online News Popularity dataset: (left) MCAR from the 16 categorical columns. (right) MCAR from randomly selected columns among 16 categorical features.

Table 2: Average RMSE with standard deviations (in parentheses) across all experiments for different imputation methods, datasets.

| Dataset | Median | GAIN | EGAIN |
|---|---|---|---|
| Breast Cancer Wisconsin | 0.0526 (0.0358) | 0.0385 (0.0331) | **0.0373** (0.0314) |
| Spambase | 0.0173 (0.0153) | 0.0200 (0.0157) | **0.0161** (0.0151) |
| Letter Recognition | 0.0666 (0.0354) | 0.0735 (0.0434) | **0.0624** (0.0357) |
| Default of Credit Card Client | 0.0417 (0.0264) | 0.0413 (0.0284) | **0.0345** (0.0263) |
| Online News Popularity | 0.0637 (0.0558) | 0.0592 (0.0548) | **0.0582** (0.0525) |

## 5 Discussion

MICE, MissForest, and GAIN are among the most commonly used methods for imputing missing values. Among them, GAIN has demonstrated superior performance by capturing complex data distributions, handling high missing rates, and learning implicit patterns without assuming predefined statistical distributions. GAIN also scales better to large datasets and adapts well to heterogeneous data types due to its adversarial training framework. However, its implementation relies on the deprecated `TensorFlow 1.x` API, incorporates several nonstandard user-defined functions, and is highly sensitive to hyperparameter tuning, often resulting in convergence issues.

In this paper, we introduced EGAIN, a modernized and enhanced version of GAIN built using `TensorFlow 2`. EGAIN addresses several of GAIN's limitations by incorporating checkpointing mechanisms for training stability, visualizations of loss functions to assist in hyperparameter selection, and standard built-in functions for initialization and scaling. Empirical results show that EGAIN consistently outperforms GAIN in terms of RMSE across multiple benchmark datasets.

EGAIN introduces key architectural improvements, most notably the use of convolutional layers in both the generator and discriminator. Although convolutional networks are traditionally applied to spatial data, recent studies have shown that they can be effective for tabular data when structural dependencies exist across features (Wang et al., 2019; Mingxun et al., 2022; Mamun & Elfouly , 2023). In EGAIN, stacking the input data and mask as separate channels enables the model to learn localized patterns between observed and missing entries, a capability that standard dense layers lack.

Despite its improved accuracy and robustness, EGAIN incurs a higher computational cost. For example, imputing 20% missing values in the credit dataset (30,000 instances) requires an average of 20 seconds using

1,000 iterations and a batch size of 256, approximately three to four times longer than GAIN. However, GAIN frequently fails to produce valid imputations at this iteration count due to instability, highlighting a key trade-off between runtime and reliability.

In summary, EGAIN offers a more stable, interpretable, and accurate solution for missing value imputation in tabular datasets. While the contributions of its individual components (e.g., convolutional layers, input reshaping, and checkpointing) were not isolated in this study, future work could explore these effects through an ablation study. Additionally, although this study focused on MCAR scenarios, follow-up research is needed to assess EGAIN's performance under MAR and MNAR mechanisms.

Table 3: Average RMSE with standard deviations (in parentheses) across 25 runs for different imputation methods, datasets, and missing data rates.

| Dataset | Missing Rate Imputation | 10% | 20% | 30% | 40% | 50% | 60% | 70% |
|---|---|---|---|---|---|---|---|---|
| Breast Cancer Wisconsin | EGAIN | **0.0210** (0.0005) | **0.0328** (0.0011) | **0.0474** (0.0037) | **0.0594** (0.0039) | **0.0743** (0.0087) | **0.0893** (0.0102) | **0.1111** (0.0056) |
| | GAIN | 0.0249 (0.0007) | 0.0372 (0.0010) | 0.0519 (0.0021) | 0.0658 (0.0010) | 0.0791 (0.0014) | 0.0940 (0.0034) | 0.1206 (0.0034) |
| | Median | 0.0433 (0.0014) | 0.0618 (0.0014) | 0.0762 (0.0014) | 0.0880 (0.0011) | 0.0986 (0.0010) | 0.1081 (0.0009) | 0.1168 (0.0009) |
| Spambase | EGAIN | **0.0147** (0.0004) | **0.0218** (0.0001) | **0.0278** (0.0000) | **0.0324** (0.0000) | **0.0368** (0.0000) | **0.0414** (0.0002) | **0.0441** (0.0001) |
| | GAIN | 0.0148 (0.0002) | 0.0219 (0.0004) | 0.0279 (0.0005) | 0.0326 (0.0004) | 0.0373 (0.0002) | 0.0416 (0.0002) | 0.0457 (0.0001) |
| | Median | 0.0165 (0.0005) | 0.0239 (0.0006) | 0.0294 (0.0007) | 0.0341 (0.0006) | 0.0382 (0.0005) | 0.0419 (0.0004) | 0.0453 (0.0004) |
| Letter Recognition | EGAIN | 0.0399 (0.0012) | 0.0575 (0.0014) | 0.0728 (0.0019) | **0.0860** (0.0026) | **0.0993** (0.0032) | **0.1111** (0.0024) | **0.1199** (0.0016) |
| | GAIN | **0.0395** (0.0011) | **0.0559** (0.0013) | **0.0703** (0.0011) | 0.0861 (0.0008) | 0.1003 (0.0023) | 0.1177 (0.0038) | 0.1460 (0.0090) |
| | Median | 0.0457 (0.0003) | 0.0646 (0.0002) | 0.0791 (0.0003) | 0.0914 (0.0002) | 0.1021 (0.0002) | 0.1120 (0.0003) | 0.1210 (0.0003) |
| Default of Credit Card Client | EGAIN | **0.0303** (0.0015) | **0.0442** (0.0018) | **0.0550** (0.0015) | **0.0632** (0.0013) | **0.0704** (0.0009) | **0.0767** (0.0017) | **0.0826** (0.0013) |
| | GAIN | 0.0304 (0.0009) | 0.0455 (0.0012) | 0.0585 (0.0018) | 0.0689 (0.0015) | 0.0787 (0.0010) | 0.0859 (0.0024) | 0.0929 (0.0014) |
| | Median | 0.0330 (0.0001) | 0.0467 (0.0002) | 0.0572 (0.0002) | 0.0661 (0.0002) | 0.0738 (0.0002) | 0.0809 (0.0001) | 0.0874 (0.0001) |
| Online News Popularity | EGAIN | 0.0552 (0.0012) | 0.0787 (0.0019) | 0.0988 (0.0015) | **0.1158** (0.0014) | **0.1300** (0.0014) | **0.1431** (0.0014) | **0.1561** (0.0041) |
| | GAIN | **0.0538** (0.0007) | **0.0772** (0.0008) | **0.0980** (0.0008) | 0.1182 (0.0007) | 0.1337 (0.0005) | 0.1475 (0.0009) | 0.1625 (0.0008) |
| | Median | 0.0624 (0.0002) | 0.0883 (0.0002) | 0.1081 (0.0002) | 0.1249 (0.0002) | 0.1396 (0.0002) | 0.1530 (0.0002) | 0.1652 (0.0002) |

Table 4: Average RMSE with standard deviations (in parentheses) across multiple runs, grouped by dataset and imputation method. Columns represent varying missing data rates (first level) and the number of missing columns (second level).

| Data | Columns | 2 | | | 8 | | |
| --- | --- | --- | --- | --- | --- | --- | --- |
| | Missing Rate Method | 25% | 50% | 75% | 25% | 50% | 75% |
| Breast Cancer Wisconsin | EGAIN | **0.0024** (0.0003) | **0.0042** (0.0004) | **0.0052** (0.0004) | **0.0107** (0.0016) | **0.0169** (0.0019) | **0.0212** (0.0026) |
| | GAIN | 0.0034 (0.0009) | 0.0061 (0.0027) | 0.0069 (0.0011) | 0.0146 (0.0005) | 0.0221 (0.0016) | 0.0281 (0.0022) |
| | Median | 0.0045 (0.0006) | 0.0063 (0.0009) | 0.0077 (0.0012) | 0.0189 (0.0016) | 0.0265 (0.0020) | 0.0325 (0.0024) |
| Spambase | EGAIN | **0.0010** (0.0000) | **0.0014** (0.0000) | **0.0016** (0.0000) | **0.0025** (0.0000) | **0.0038** (0.0000) | **0.0047** (0.0000) |
| | GAIN | 0.0010 (0.0000) | 0.0015 (0.0001) | 0.0020 (0.0006) | 0.0027 (0.0002) | 0.0041 (0.0003) | 0.0052 (0.0004) |
| | Median | 0.0010 (0.0004) | 0.0014 (0.0005) | 0.0017 (0.0005) | 0.0035 (0.0007) | 0.0050 (0.0009) | 0.0062 (0.0011) |
| Letter Recognition | EGAIN | **0.0075** (0.0012) | **0.0101** (0.0017) | **0.0127** (0.0023) | **0.0312** (0.0024) | **0.0453** (0.0027) | **0.0565** (0.0045) |
| | GAIN | 0.0085 (0.0009) | 0.0118 (0.0015) | 0.0160 (0.0020) | 0.0330 (0.0025) | 0.0483 (0.0010) | 0.0690 (0.0042) |
| | Median | 0.0088 (0.0008) | 0.0123 (0.0013) | 0.0150 (0.0016) | 0.0364 (0.0013) | 0.0515 (0.0018) | 0.0631 (0.0022) |
| Default of Credit Card Client | EGAIN | **0.0038** (0.0001) | **0.0051** (0.0001) | **0.0063** (0.0002) | **0.0111** (0.0002) | **0.0155** (0.0002) | **0.0193** (0.0005) |
| | GAIN | 0.0043 (0.0004) | 0.0064 (0.0011) | 0.0083 (0.0020) | 0.0130 (0.0007) | 0.0190 (0.0013) | 0.0240 (0.0027) |
| | Median | 0.0045 (0.0036) | 0.0064 (0.0051) | 0.0078 (0.0062) | 0.0180 (0.0058) | 0.0255 (0.0082) | 0.0312 (0.0101) |
| Online News Popularity | EGAIN | **0.0027** (0.0015) | **0.0039** (0.0022) | **0.0048** (0.0027) | **0.0114** (0.0025) | **0.0164** (0.0033) | **0.0200** (0.0044) |
| | GAIN | 0.0029 (0.0015) | 0.0047 (0.0022) | 0.0062 (0.0030) | 0.0132 (0.0031) | 0.0171 (0.0042) | 0.0215 (0.0049) |
| | Median | 0.0032 (0.0019) | 0.0045 (0.0027) | 0.0055 (0.0033) | 0.0134 (0.0030) | 0.0190 (0.0042) | 0.0232 (0.0052) |

| Data | Columns | 14/16[*] | | |
| --- | --- | --- | --- | --- |
| | Missing Rate Method | 25% | 50% | 75% |
| Breast Cancer Wisconsin | EGAIN | **0.0207** (0.0018) | **0.0339** (0.0029) | **0.0463** (0.0031) |
| | GAIN | 0.0262 (0.0007) | 0.0395 (0.0017) | 0.0488 (0.0014) |
| | Median | 0.0368 (0.0020) | 0.0521 (0.0026) | 0.0639 (0.0032) |
| Spambase | EGAIN | **0.0052** (0.0001) | **0.0079** (0.0000) | **0.0099** (0.0001) |
| | GAIN | 0.0054 (0.0001) | 0.0081 (0.0002) | 0.0100 (0.0001) |
| | Median | 0.0070 (0.0008) | 0.0101 (0.0010) | 0.0124 (0.0013) |
| Letter Recognition | EGAIN | **0.0563** (0.0023) | **0.0855** (0.0023) | **0.1067** (0.0041) |
| | GAIN | 0.0567 (0.0020) | 0.0865 (0.0027) | 0.1532 (0.0141) |
| | Median | 0.0632 (0.0011) | 0.0893 (0.0015) | 0.1094 (0.0019) |
| Default of Credit Card Client | EGAIN | **0.0163** (0.0004) | **0.0231** (0.0004) | **0.0286** (0.0005) |
| | GAIN | 0.0196 (0.0008) | 0.0283 (0.0011) | 0.0338 (0.0016) |
| | Median | 0.0311 (0.0064) | 0.0441 (0.0091) | 0.0540 (0.0111) |
| Online News Popularity | EGAIN | **0.0224** (0.0035) | **0.0328** (0.0050) | **0.0397** (0.0066) |
| | GAIN | 0.0256 (0.0045) | 0.0349 (0.0035) | 0.0433 (0.0076) |
| | Median | 0.0263 (0.0043) | 0.0372 (0.0061) | 0.0456 (0.0074) |

[*] 14 for Letter Recognition, Default of Credit Card Client; 16 for Breast Cancer Wisconsin, Spambase, Online News Popularity.

## Abbreviations

| | |
|---|---|
| GAIN | Generative Adversarial Imputation Network |
| EGAIN | Enhanced Generative Adversarial Imputation Network |
| MAR | Missing At Random |
| MCAR | Missing Completely At Random |
| MNAR | Missing Not At Random |
| MICE | Multiple Imputation by Chained Equations |
| LFM-D2GAIN | Latent Factor Model with Dual Discriminator GAIN |
| GAGIN | Generative Adversarial Guider Imputation Network |
| ccGAIN | Conditional Clinical GAIN |
| LWGAIN | Loss Wasserstein GAIN |
| scGAIN | single-cell GAIN |
| CGAIN | Conditional GAIN |
| TensorFlow | TF |
| API | Application Programming Interface |
| RMSE | Root Mean Squared Error |

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
