# OpenReview forum: "EGAIN: Enhanced Generative Adversarial Networks for Imputing Missing Values"
_TMLR — Rejected by TMLR_

### Review · Reviewer_PRqL · 2025-05-18

**Summary Of Contributions:**

The authors provide several functions to the code of a missing value imputation methods, GAIN. The contributions is mostly engineering-level.

**Audience:**

Yes

**Claims And Evidence:**

Yes

**Requested Changes:**

1. Clear description of training and test

2. Solid ablation with explanation on EGAIN

3. A non-GAN baseline

**Strengths And Weaknesses:**

Strengths:

The authors provide many functions to the original code of GAIN, including tf version update, more complex networks and more.

Weaknesses:

1. I do not find where the authors define the training/test datasets. GAIN uses a cross-validation method. Does EGAIN also use it? The RMSE improvement is so high. There is a chance that the model overfits to the test dataset.

2. It is not clear that which part of EGAIN improves the RMSE. The author should do an ablation study on this to make the results more convincing. It is also not clear to me why deep convolutional networks are better on tabular data settings.

3. GAN training is highly unstable. The authors should include a non-GAN baseline to make a fair comparison.

---

> ### Author Response · Authors · 2025-05-27
> **Response to Reviwer Comments**
>
> Thank you for your thoughtful comments. We would like to first bring to your attention the GAIN process, followed by our point to pint response. For each iteration (not epoch) a minibatch (batch size) of data with missing values is selected randomly without replacement. Using the minibatch, first the generator and then the discriminator provide their estimation of that batch. Then, a loss function is calculated for their performance and network weights are adjusted to minimize their loss. This completes a training cycle, and the next iteration starts which follows the same process. Once the set number of iterations are complete, the generator operates on the entire data with missing and provides imputations.
>
> 1.
> GAIN (Yoon et al., 2018) conducts imputation experiments using a 5-fold cross-validation procedure, repeated 10 times. The reported performance metrics, including the average RMSE and standard deviation, are calculated by comparing the imputed values to the original data across these runs.
>
> In our study, we adopt a procedure that more closely mirrors real-world missing data scenarios. Specifically, missing values are introduced into the complete datasets using a missingness-generating function with different random seeds. These incomplete datasets are then passed to the EGAIN (or GAIN) algorithm for imputation. Importantly, the model does not have access to the ground truth during training or inference, ensuring a fair evaluation setting. Hyperparameters for each model were selected using 5-fold cross-validation, but the imputation itself was performed on the full dataset with synthetic missingness. The final RMSE is computed by comparing the imputed values to the known ground truth, and average RMSE and standard deviation are reported across 25 independent runs. Identical random seeds were used for both GAIN and EGAIN to ensure comparability.
>
> In summary, while we do not follow the exact 5-fold CV-based evaluation scheme described in the original GAIN paper, we use a practically motivated strategy that (1) avoids data leakage, (2) allows consistent hyperparameter tuning, and (3) reflects how imputation is typically performed in real-world applications the entire incomplete dataset is passed to the imputation function.
>
> The materials section of the manuscript is revised to better explain the applied process.
>
> 2.
> We appreciate the reviewer’s insightful comment regarding the use of convolutional networks for tabular data and the request to clarify which components of EGAIN drive the performance improvements.
>
> We acknowledge that convolutional architectures are traditionally applied to spatial data such as images, and their application to tabular data may not seem intuitive at first glance. However, recent studies (e.g., Wang et al., 2019; Mingxun et al., 2022; Mamun & Elfouly, 2023) have demonstrated that CNNs can be effective for tabular tasks when structural correlations exist across features. In EGAIN, we stack the input data and the corresponding mask array as separate channels and apply Conv1D layers. This design allows the network to learn localized patterns between observed and missing entries, which are harder to capture using fully connected layers. As a result, the model can better exploit dependencies across neighboring features, leading to more plausible imputations.
>
> Regarding the performance gains, we emphasize that the improvement in RMSE is the result of several cumulative enhancements. Specifically:
> 1. Switching from dense to convolutional layers, which improves pattern recognition in the presence of structured missingness;
> 2. Restructuring the input format by stacking data and mask arrays, which enhances the model’s capacity to learn missingness patterns. Our prior experiential results showed that this step works only in conjunction with step 1 and does not grant improvement alone to the GAIN process;
> 3. Incorporation of checkpointing and training stability mechanisms, which address the convergence issues frequently observed in GAIN;
>
> We invite the reviewer to refer to the supplementary Excel file, which details the hyperparameters used and the RMSE scores obtained for each run of GAIN and EGAIN. These results also illustrate one key limitation of GAIN: the tendency to fail under poor hyperparameter settings. In contrast, EGAIN includes visual tools (e.g., loss charts) to assist in hyperparameter tuning and improves overall training robustness.
>
> We have revised the discussion section of the manuscript to clarify the motivation for using convolutional layers in tabular settings and provided additional references and discussion on their suitability. We hope this addresses the reviewer’s concerns and helps clarify the contributions of each component in EGAIN’s architecture.
>
> 3.
> Thank you for this constructive suggestion. We are working on adding a baseline imputation method to further compare imputation performance of EGAIN.

---

> > ### Comment · Reviewer_PRqL · 2025-06-16
> > **Reviewer Response**
> >
> > Thanks for the responses. I expect to see a new version of the paper that addresses all my questions. Currently, the paper is not in a good shape. Could you do that first? Also, for 3, what are the results for the baseline imputation method?

---

> > > ### Author Response · Authors · 2025-06-18
> > > **Response to Reviwer Comments**
> > >
> > > Thank you for your constructive and insightful feedback, which has greatly contributed to improving the quality of the manuscript. In response to the received comments, we have made several significant revisions, including:
> > >
> > > 1. Inclusion of Median Imputation as a Baseline: We have added Median imputation as a baseline method to benchmark the performance of both GAIN and EGAIN. The results demonstrate statistically significant improvements over this baseline.
> > >
> > > 2. Expanded Dataset Evaluation: We clarified the use of real-world benchmark datasets (Breast Cancer Wisconsin, Spambase, and Credit Card Default), and we have extended our evaluation by including two additional datasets, Letter Recognition and Online News Popularity, that were used in the original GAIN study. These additions provide a more comprehensive assessment of all three methods.
> > >
> > > 3. Revised Introduction: The Introduction has been revised to clearly articulate the key enhancements introduced by EGAIN, with a specific focus on distinguishing our architectural and training improvements from those of the original GAIN implementation.
> > >
> > > 4. Improved Methods Section: The "Materials and Methods" section has been reorganized and clarified to better present EGAIN’s innovations and provide a more transparent description of the experimental setup.
> > >
> > > 5. Updated Discussion: We have revised the Discussion section to include a dedicated explanation of the motivation and rationale for using convolutional layers in tabular data settings.
> > >
> > > All changes have been marked in blue in the revised manuscript to facilitate review and tracking.

---

> > > > ### Comment · Reviewer_PRqL · 2025-06-19
> > > > **Where is the revised manuscript**
> > > >
> > > > I am sorry. Where is the revised manuscript? I clicked the pdf button and did not find any change.

---

> > > > > ### Author Response · Authors · 2025-06-19
> > > > > **Revised Manuscipt**
> > > > >
> > > > > Thank you for letting us know of the issue. The revised manuscript is now shared with the reviewers where all changes have been marked in blue.

---

> > > > > > ### Comment · Reviewer_PRqL · 2025-06-27
> > > > > > **Thanks for the updates**
> > > > > >
> > > > > > I thank the authors for the updates. I do not have further questions. I also do not think novelty should be a metric for this journal as other reviewers said. I tend to accept the paper.

---

### Review · Reviewer_enHA · 2025-05-27

**Summary Of Contributions:**

This paper deals with imputing missing values and proposes to revisit the GAIN paper.
- The main contribution of the paper is an updated implementation of the original GAIN paper with the following changes :
	- Re-implemented GAIN in the updated Tensorflow version.
	- Changed the network architecture of the GAIN from a fully-connected layer to a convolutional generator.
	- Changed scaling and activation function to standard built-in functions.
	- Provides a chart displaying the loss and a visual indication of performance.
- This implementation is evaluated on a subset of the moderate size datasets from UCI (used in the original GAIN paper) and compared to the original GAIN paper implementation.

**Audience:**

No

**Claims And Evidence:**

No

**Requested Changes:**

In the current state, the paper's contributions are too minor (only re-implementation of existing work) for acceptance.
It could be extended by a proper empirical study of the performance of GAIN compared to other papers and datasets. For instance, one could extend the implementation to all the different existing GAIN extensions (Wasserstein GAINs ...) and carefully compare their performance. I would also recommend extending this study to other existing modern deep generative models for missing data (MIWAE, MissDiff ...). Finally, real datasets should be considered for the study to strengthen the paper.

**Strengths And Weaknesses:**

- Strengths:
	- Updated implementation of GAIN.
	 - Carefully described experiments with all hyperparameters described.
	 - Carefully described performance with statistical significance (p-values)
- Weaknesses :
	- The main contribution of the papers is the implementation. In such a case, the implementation should have also been given for review in the supplementary.
	- The implementation itself is straightforward, with only minor transformation / re-implementation of the original paper. For instance, it seems that the main contribution of the paper is the transformation of the fully connected neural network into a convolutional neural which is a fairly standard practice in ML and can't be considered a major contribution.
	- Though the method seems to perform better on the three provided datasets, these datasets and associated missingness patterns are very simple and perhaps should be extended to real datasets and real missing patterns.  The very least should be to extend this study to all the datasets tested in the original GAIN papers.
	- The review of existing work is incomplete with no mention of other modern DeepGenerative approach (VAE based, Diffusion based ...)
	- The fact that GAIN is better than the other more traditional methods (not based on DeepLearning) is still up to discussion, as such, comparison with such methods would be necessary with this new implementation (comparison to MICE, MissForest).
- Claim issues :
	- The authors mentioned that EGAIN worked consistently in MAR and MNAR scenarios, but this was neither shown empirically nor theoretically in the paper. Without evidence to back that claim, this should be taken out of the paper, especially since it is one of the most standard critiques for GAIN [1].
	- It is mentioned that GAIN "often failed to generate results when the number of iterations exceeded an optimal selected threshold," but this was not properly described or proven.

[1]Sun, Y., Li, J., Xu, Y., Zhang, T., & Wang, X. (2023). Deep learning versus conventional methods for missing data imputation: A review and comparative study. _Expert Systems with Applications_, _227_, 120201.

---

> ### Author Response · Authors · 2025-05-29
> **Response to Reviwer Comments**
>
> We thank the reviewers for their careful reading of our manuscript. Our responses to the comments are organized under the following categories:
>
> A) Real Datasets
> 1. The Breast Cancer Wisconsin [https://archive.ics.uci.edu/dataset/17/breast+cancer+wisconsin+diagnostic], Default of Credit Card Client [https://archive.ics.uci.edu/dataset/350/default+of+credit+card+clients], and Spambase [https://archive.ics.uci.edu/dataset/94/spambase] datasets are all real-world datasets that have been widely adopted as benchmarks in prior work, including in GAIN (Yoon et al., 2018), GAGIN (Wang et al., 2022), and ClueGAIN (Zhao et al., 2023), among others.
> 2. When working with real datasets that contain missing values, it is typically impossible to verify the true values of the missing entries. As a result, the performance of imputation methods cannot be validated directly on such datasets. To address this limitation, a standard simulation approach is used across the literature: missing values are synthetically introduced into complete datasets using established mechanisms (e.g., MCAR), and the imputed values are then compared against the original known values to calculate performance metrics. This methodology has been used in numerous studies (including those cited above) to evaluate imputation accuracy in a controlled setting.
> 3. Consistent with this standard practice, we have used three real-world benchmark datasets that represent different scales [Breast Cancer (small), Spambase (medium), and Credit (large)] to simulate missingness and assess the performance of our model. We have revised the manuscript to better reflect these points.
>
>
> B) Comparisons
> 1. Many GAIN successors do not have publicly available codebases, including D2GAIN (Shen et al., 2022), GAGIN (Wang et al., 2022), ClueGAIN (Zhao et al., 2023), ccGAN (Bernardini et al., 2023), and LWGAIN (Qian et al., 2024, a Wasserstein-based variant). Some do not have publicly available datasets.
> 2. Among those that do share their code, such as scGAIN (Gunady et al., 2019), COGAIN (Dong et al., 2021), and CGAIN (Awan et al., 2023), the implementations rely on the outdated TensorFlow 1.x API. Since TensorFlow 2 introduced major improvements in usability, execution, and integration with Keras, the transition to TF2 has become a necessity for modern, robust implementations. EGAIN is implemented entirely in TF2, with significant structural changes to the data pipeline, model architecture, and optimization procedures. It also addresses a common limitation in GAIN (training instability) by incorporating checkpointing mechanisms and diagnostic visualizations for both the generator and discriminator loss curves.
> 3. The original GAIN codebase, available only via GitHub, often requires non-trivial troubleshooting to function properly. In contrast, we provide both pip and GitHub installable versions of EGAIN, making it easy to deploy with a simple three-line script.
> 4. While EGAIN introduces several architectural and usability improvements over GAIN, we do not claim it to be superior to every existing imputation method. Rather, we present it as a practical and enhanced alternative to GAIN. As an additional reference point, we have included results from a baseline median imputation method in the revised manuscript.
>
> C) Claims
> 1. We invite the reviewer to examine the supplementary files, where multiple instances of GAIN failing to complete imputations are documented. Upon investigation of the original source code, we identified that such failures are often caused by instability in the generator's optimization process. When divergence occurs at any iteration, the training process halts entirely, preventing further progress. These failures are especially likely at higher iteration counts. Notably, these convergence issues were less frequent in the credit dataset. Our observations are consistent with findings from previous studies, including Kazemi (2020) and Qin et al. (2024), which also report instability in GAIN under certain settings.
> 2. We would like to clarify that our manuscript does not make any claims regarding EGAIN’s performance under MAR or MNAR scenarios. The only related statement is in the last paragraph of the Discussion section, which reads: “This study focused solely on missing value imputation in MCAR scenarios. A follow-up study is required to compare the performance of EGAIN and its alternatives in MAR and MNAR scenarios.”
> This reflects our intent to limit conclusions to the MCAR setting and acknowledges the need for future research on other missingness mechanisms.

---

> > ### Comment · Reviewer_enHA · 2025-06-02
> > **Response to Authors**
> >
> > I want to thank the authors for their detailed responses and the extension to a new baseline (median imputation). However, my concerns regarding the **novelty of the paper** and the **extensiveness of the experimental evaluation** remain unaddressed.
> >
> > ### A) Real Datasets:
> >
> > - 1. While the selected datasets have indeed been used in prior work, evaluating the proposed method on only three datasets, with comparison solely with the original GAIN implementation, is insufficient to demonstrate the effectiveness of E-GAIN. Furthermore, the original GAIN paper evaluated performance on two additional datasets (_News_ and _Letter_). Is there a specific reason these were omitted from your evaluation?
> > - 2. In the context of missing data, it is also common practice to assess the quality of imputations via downstream performance (e.g., classification accuracy on imputed datasets, as done in MIWAE) or by qualitatively comparing imputations on test distributions (e.g., propensity score estimation in Not-MIWAEs). These types of evaluations would help better support the claims of improved imputation quality.
> > - 3. Would it be possible to clarify the 'scale' of the datasets? In terms of dimensions, the 3 datasets are fairly medium-sized.
> >
> > ### B) Comparisons:
> >
> > - 2. The stated limitation regarding checkpointing and diagnostic visualisation is not specific to GAIN, but rather a general challenge in most machine learning implementations. While such practices are certainly useful, they are standard across the field and do not, in themselves, represent a substantial methodological advancement.
> >
> > ### C) Claims:
> >
> > - 1. While it is appreciated that failure cases are included in the supplementary materials as tabular data, these results should be included in the main text, preferably in the form of a dedicated paragraph or a well-designed figure. This would more clearly convey the frequency and severity of failures observed with the original GAIN implementation.
> > - 2. I was indeed misled by the paragraph on page 3, which sequentially discusses MAR and MNAR scenarios and then the relative performance of E-GAIN:
> >     > “GAIN implementation is highly sensitive to hyperparameter selection and may fail to converge or produce results if the number of iterations is not appropriate. This issue is particularly evident when missing values exist in only a few columns, as seen in MAR and MNAR scenarios. In contrast, EGAIN consistently provides reliable imputations in every run.”
> >
> > This phrasing implies that E-GAIN is effective in MAR and MNAR settings, though this paper does not appear to rigorously evaluate that claim. It would be beneficial to clarify in the main text that such scenarios are not explicitly addressed in this work.

---

> > > ### Author Response · Authors · 2025-06-04
> > > **Response to Reviwer Comments**
> > >
> > > We are grateful to the reviewer for their thorough, constructive, and insightful feedback, which has significantly improved the quality, clarity, and rigor of our manuscript.
> > >
> > > A) Real Datasets
> > >
> > > 1. In response to the reviewer’s valuable suggestion, we have expanded our evaluation to include two additional datasets (Letter Recognition and Online News Popularity) which were also used in the original GAIN paper (Yoon, 2018). These datasets are now included in our experiments comparing EGAIN against both the original GAIN implementation and baseline Median imputation.
> > > 2. While downstream tasks (e.g., classification accuracy) offer valuable insights, our evaluation focuses on Root Mean Squared Error (RMSE), which remains the standard metric used in many published works, including the original GAIN paper, for assessing imputation accuracy in datasets with mixed variable types. RMSE allows for a direct and quantitative comparison of imputed values against the ground truth in simulation studies. Given the broad range of datasets, missingness scenarios, and comparative baselines, we believe our results robustly support the performance improvements introduced by EGAIN.
> > > 3. The number of cases and features in each dataset is already mentioned in Table 1 of the manuscript. Our selection for evaluation strategy was to pick one from each small, medium, and large datasets with a mix of continuous and discrete features. Given the relative scale of each dataset, this lead to selection of Breast Cancer Wisconsin (n=569), Spambase (n=4,601), and Default of Credit Card Client (n=30,000) as small, medium, and large. Furthermore, as stated in our response, we have included performance comparisons for Letter Recognition (n=20,000) and Online News Popularity (n=39,797) in the revised manuscript.
> > >
> > > B) Comparisons
> > >
> > > Since GAIN was introduced in 2018 using TensorFlow 1.x, and TensorFlow 2 was released in 2019, many subsequent papers have continued using or extending the outdated GAIN codebase. In contrast, EGAIN is implemented in TensorFlow 2.x, fully supports pip-based installation, and includes a GitHub repository with example runs, documentation, and diagnostics tools, making it more accessible, maintainable, and easier to integrate into contemporary ML pipelines.
> > > Following the reviewers’ feedback, we have revised the “Materials and Methods” section to more clearly distinguish between:
> > > •	Methodological improvements (e.g., architectural changes that address convergence issues), and
> > > •	Engineering enhancements (e.g., standardized function replacement, modular design, TensorFlow 2 compatibility).
> > >
> > > C) Claims
> > >
> > > 1. We appreciate the reviewer’s suggestion. The data in the supplementary material are details of each simulation run, mainly provided to show how frequent GAIN fails to impute missing values. The average across simulations of this data is already presented in the paper in Figure 1, 3, 4, and Table 2. We have provided explanations to the revised manuscript for clarity.
> > > 2. We understand the concern of the reviewer and revised the manuscript to better reflect the study. Furthermore, our simulations consistently showed that GAIN fails more frequently when missingness is limited to a small subset of variables, which aligns with the structural characteristics often observed in MAR/MNAR settings. However, we now explicitly note that our study does not empirically evaluate these mechanisms, and we frame this as a limitation and future direction.

---

> ### Author Response · Authors · 2025-05-29
> **References**
>
> D) References
> \bibitem[Yoon et al.(2018)]{GAIN}
> Yoon, J.; Jordon, J.; Van Der Schaar, M. {GAIN}: Missing Data Imputation using Generative Adversarial Nets. In Proceedings of the 35th International Conference on Machine Learning; Stockholm, Sweden, July 10-15, 2018.
>
> \bibitem[Wang et al.(2022)]{GAGIN}
> Wang, W.; Chai, Y.; Li, Y. GAGIN: generative adversarial guider imputation network for missing data. {\em Neural Computing and Applications} {\bf 2022}, {\em 34}, 7597--7610.
>
> \bibitem[Zhao(2023)]{ClueGAIN}
> Zhao, S. ClueGAIN: Application of Transfer Learning On Generative Adversarial Imputation Nets (GAIN). {\em arXiv} {\bf 2023}.
>
> \bibitem[Gunday et al.(2019)]{scGAIN}
> Gunady, M.K., Kancherla, J., Bravo, H.C., Feizi, S. scGAIN: Single Cell RNA-seq Data Imputation using Generative Adversarial Networks, BioArchive, 2019.
>
> \bibitem[Shen et al.(2022)]{shen22}
> Shen, Y.; Zhang, C.; Zhang, S.; Yan, J.; Bu, F. LFM-D2GAIN: An Improved Missing Data Imputation Method Based on Generative Adversarial Imputation Nets. In Proceedings of the 2022 IEEE International Conference on Electrical Engineering, Big Data and Algorithms (EEBDA); Changchun, China, February 25-27, 2022.
>
> \bibitem[Bernardini et al.(2023)]{ccGAIN}
> Bernardini, M.; Doinychko, A.; Romeo, L.; Frontoni, M.; Amini, M. A novel missing data imputation approach based on clinical conditional Generative Adversarial Networks applied to EHR datasets. {\em Computers in Biology and Medicine} {\bf 2023}, {\em 163}, 107188.
>
> \bibitem[Qian et al.(2024)]{LWGAIN}
> Qian, H.; Geng, Y.; Wang, H.; Wu, X.; Li, M. LWGAIN: An Improved Missing Data Imputation Method Based on Generative Adversarial Imputation Network. In Proceedings of the 2024 5th International Conference on Computer Vision, Image and Deep Learning (CVIDL); Zhuhai, China, April 19-21, 2024.
>
> \bibitem[Dong et al.(2021)]{COGAIN}
> Dong, W.; Fong, D.; Yoon, J.; Wan, E.; Bedford, L.; Tang, E.; Lam, C. Generative adversarial networks for imputing missing data for big data clinical research. {\em BMC Medical Research Methodology} {\bf 2021}, {\em 21}, 78.
>
> \bibitem[Awan et al.(2020)]{CGAIN}
> Awan, S.E., Bennamoun, M., Sohel, F., Sanfilippo, F.M., Dwivedi, G. Imputation of Missing Data with Class Imbalance using Conditional Generative Adversarial Networks. {\em Archive} {\bf 2020}.
>
> \bibitem[Qin et al.(2024)]{MGAIN}
> Qin, X., Shi, H., Dong, X., Zhang, S., Yuan, L. Improved generative adversarial imputation networks for missing data. {\em Applied Intelligence} {\bf 2024}, {\em 54(21)}, 11068-11082.
>
> \bibitem[Kazemi et al.(2020)]{IGANI}
> Kazemi, A., Meidani, H. IGANI: Iterative Generative Adversarial Networks for Imputation with Application to Traffic Data. {\em Archive} {\bf 2020}.

---

### Review · Reviewer_X2ak · 2025-05-30

**Summary Of Contributions:**

Authors focus on the broad domain of missing values and the task of data imputation, which is common in several applications, such as medical. They propose the Enhanced Generative Adversarial Network (EGAIN), which builds upon the commonly used Generative Adversarial Imputation Network (GAIN) method and provide improvements with respect to its convergence issues (via loss monitoring, hyperparameter tuning, weight initialization), architectural design (by replacing dense layers with convolution) and implementation code (upgrading to newer python and tensorflow versions). Results on 3 UCI datasets showcase performance improvements in terms of the RMSE metric.

**Audience:**

No

**Claims And Evidence:**

No

**Requested Changes:**

Based on the above, the authors could:
- Reformulate the theoretical and methodological contributions of their work in the introduction and the methods section (via equations, figures), such that it becomes clear which parts they advance and how.
- Clearly position their work and contribution against the most recent and competitive in the field - this necessitates a more extended related works section.
- Extend the experimental evaluation against more baselines/datasets to prove consistent performance improvement.

**Strengths And Weaknesses:**

**Strengths:**
The authors propose practical solutions to a framework that seems quite standard for practitioners in handling datasets with missing data.

**Weaknesses:**
- The proposed contribution lacks proper theoretical presentation, mathematical justifications, and rather seems like an incremental, very technically-focused improvement of a well-established framework.
- The contributions of the work are poorly presented in the abstract and the introduction section, and relevant sentences about contributions lack specificity.
- The experimental evaluation is limited to comparing the proposed EGAIN method to only its closely related GAIN, while more recent related methods described in the introduction (such as ClueGAIN, ccGAIN) are not experimentally compared as baselines.
- An extended related works section is missing, failing to position the proposed method against existing methods in the literature.
- The evaluated datasets are a subset of the ones presented in GAIN (which also does not represent the most recent publication in the domain). It would be important that the authors showcase their performance improvements with a more extended evaluation (more datasets, ablation studies against GAIN).
- The field of data imputation is quite general and usually extends to time series data and other tasks subsequent such as classification and forecasting. For instance, in the time series domain, there are several very recent deep learning methods that have shown very robust performances. It is unclear to what data types/dynamics (e.g., continuous/discrete) and tasks the proposed method extends.

---

> ### Author Response · Authors · 2025-06-02
> **Response to Reviwer Comments**
>
> A) We appreciate the reviewer’s suggestion to clarify the theoretical and methodological contributions of our work. In response, we have:
>
> •	Revised the final paragraph of the Introduction to explicitly highlight the main enhancements introduced by EGAIN, distinguishing our architectural and training improvements from the prior GAIN method.
>
> •	Revised the Materials and Methods section to present EGAIN’s innovations in a clearer format.
>
> These changes aim to make our contributions more transparent and theoretically grounded.
>
> B) Thank you for this valuable recommendation. We have expanded the Related Work section to better situate EGAIN within the current landscape of GAIN-based imputation methods. The updated section now:
>
> •	Includes discussion of recent GAIN variants, such as D2GAIN, GAGIN, ClueGAIN, ccGAIN, and LWGAIN, outlining their primary contributions and limitations.
>
> •	Compares EGAIN’s key design choices (e.g., use of CNNs, input structuring, convergence solutions) with those of existing approaches, highlighting areas where EGAIN offers improvements.
>
> •	Clarifies the novelty of EGAIN, particularly in terms of architectural simplicity, training stability, and practical usability (e.g., TF2.x implementation, pip-installable code).
>
> These additions aim to provide a more complete view of recent advances while clearly distinguishing our contributions.
>
> C) We appreciate the reviewer’s emphasis on robust evaluation. In the revised manuscript, we have:
>
> •	Added an additional baseline: Median imputation, to contextualize EGAIN’s performance relative to a simple and widely-used non-learning-based approach.
>
> •	Clarified the use of real benchmark datasets (Breast Cancer Wisconsin, Spambase, and Credit Card Default), and are working to include News Dataset to the list of comparisons to be consistent with prior work on GAIN.
>
> We believe these additions strengthen the empirical support for EGAIN and appreciate the reviewer’s encouragement to improve the experimental rigor.

---

### Decision · Action_Editor_Rakd · 2025-07-22

**Recommendation:** Reject

**Audience:**

No

**Audience Explanation:**

While the reviewers acknowledged and appreciated the authors' efforts to improve the manuscript, including the addition of a median imputation baseline, an expanded set of datasets, and revised sections, the majority of the reviewers still view its main contribution as a reimplementation of the original GAIN paper using techniques that are already widely adopted in the field.

**Claims And Evidence:**

No

**Claims Explanation:**

While some reviewers were satisfied that all claims made in the paper were supported by the evidence, others expressed concerns that the claims regarding the method's robustness and advantages were not convincingly substantiated and required further experimental validation, including additional baselines and datasets.